# Knowledge, Attitudes, Perceptions and Vaccination Acceptance/Hesitancy among the Community Pharmacists of Palermo’s Province, Italy: From Influenza to COVID-19

**DOI:** 10.3390/vaccines10030475

**Published:** 2022-03-18

**Authors:** Claudio Costantino, Giorgio Graziano, Nicole Bonaccorso, Arianna Conforto, Livia Cimino, Martina Sciortino, Francesco Scarpitta, Chiara Giuffrè, Salvatore Mannino, Mario Bilardo, Caterina Ledda, Francesco Vitale, Vincenzo Restivo, Walter Mazzucco

**Affiliations:** 1Department of Health Promotion Sciences, Maternal and Infant Care, Internal Medicine and Medical Specialties (PROMISE) “G. D’Alessandro”, University of Palermo, 90127 Palermo, Italy; giorgio.graziano@unipa.it (G.G.); nicole.bonaccorso@unipa.it (N.B.); arianna.conforto@unipa.it (A.C.); livia.cimino@unipa.it (L.C.); martina.sciortino@unipa.it (M.S.); francesco.scarpitta@unipa.it (F.S.); francesco.vitale@unipa.it (F.V.); vincenzo.restivo@unipa.it (V.R.); walter.mazzucco@unipa.it (W.M.); 2Order of Pharmacists of Palermo Province, 90146 Palermo, Italy; farmacia.chiaragiuffre@gmail.com (C.G.); farmaciamannino@gmail.com (S.M.); mbilardopa@gmail.com (M.B.); 3Occupational Medicine, Department of Clinical and Experimental Medicine, University of Catania, 95100 Catania, Italy; caterina.ledda@unict.it

**Keywords:** influenza vaccination, COVID-19 vaccination, community pharmacists, KAPs, vaccine hesitancy, healthcare professionals, SARS-CoV-2 infection

## Abstract

In Italy, following the start of the SARS-CoV-2 vaccination campaign, community pharmacies (CPs) were recruited on a voluntary basis in order to administer COVID-19 vaccines as part of their activities. The aim of the present study was to investigate the knowledge, attitudes, and practices regarding SARS-CoV-2 infection prevention, and vaccine acceptance/hesitancy towards COVID-19 and influenza vaccinations among the community pharmacists operating in the Palermo Province. A cross-sectional study was conducted, with two different questionnaires administered before and after the conduction of the vaccination campaign against SARS-CoV-2 at the COVID-19 vaccination center of the Palermo University Hospital (PUH). The baseline survey showed that 64% of community pharmacists (CPs) declared that they planned to vaccinate against SARS-CoV-2, and 58% were vaccinated against influenza during the 2020/2021 season. Factors significantly associated with willingness to receive the COVID-19 vaccination were confidence in vaccines (adjOR 1.76; CI 1.11–2.80), fear of contracting SARS-CoV-2 infection (adjOR 1.50; CI 1.06–2.11), considering COVID-19 vaccination to be the best strategy to counteract SARS-CoV-2 (adjOR 1.79; CI 1.39–2.29), and adherence to influenza vaccination during the 2020/2021 season (adjOR 3.25; CI 2.23–4.25). The adherence among CPs of the Palermo Province to COVID-19 vaccination was 96.5%. From the post-vaccination survey, the main reasons for changing opinions on vaccination adherence were the introduction of mandatory vaccinations, fear of contracting COVID-19, and limitations on work activities in the case of vaccine refusal. The achievement of very high COVID-19 vaccination coverage rates among healthcare professionals (HCPs) in the present study was mainly due to the mandatory vaccination policies; nevertheless, a willingness for COVID-19 vaccination was relatively high among pharmacists before the beginning of the vaccination campaign. HCPs and CPs should receive training on vaccination, which is recommended in the national immunization plan and is also suggested by the respondents in our study, in order to routinely re-evaluate their own vaccination profiles, as well as those of their patients.

## 1. Introduction

Community pharmacies (CPs) have been identified as accessible venues that provide preventive services, and a general agreement was documented on their positive impact in increasing access to preventive health services, particularly among individuals who otherwise would not be reached by other healthcare providers [1].

In the United States (US), for over a decade, CPs have been involved in promoting a culture of vaccination, encouraging patients to receive necessary vaccinations, and administering the vaccines themselves [2]. The data obtained from subsequent studies have amply demonstrated how pharmacists have contributed to increasing vaccination coverage. Thanks to the widespread distribution of pharmacies throughout the territory, they have managed to reach even those who otherwise would not have had the opportunity to be vaccinated [3,4,5,6,7,8,9,10,11]. In the US, pharmacies have become the second most popular place to receive influenza vaccination, after the offices of general practitioners (GPs) [12].

In Canada, the active involvement of pharmacists in the anti-influenza campaign has ensured a progressive increase in vaccination coverage, taking advantage of the counseling activity and the climate of trust that is usually established between the patient and the reference pharmacy [13,14,15,16].

In several European countries (France, Ireland, Portugal, United Kingdom, Hungary, and Poland), pharmacies are actively involved during vaccination campaigns, especially for anti-influenza vaccine administration [17,18].

Community pharmacies participate in various programs of prevention and health promotion in Italy, adapting their methodologies to the structures of where they operate, and providing ad hoc training courses for their healthcare workers when necessary [19,20,21,22].

The current Italian National Vaccination Plan (NVP) aims to promote the improvement of the knowledge and attitudes of healthcare professionals (HCPs) on preventive strategies against vaccine preventable diseases (VPDs), while disseminating a culture of vaccination among the general population [23,24]. As already stated, for general practitioners (GPs), community pharmacists may represent a reference point for the general population regarding health and preventive issues such as vaccines and vaccinations [19,25].

In Italy, following the start of the SARS-CoV-2 vaccination campaign on 29 March 2021, an agreement for the administration of the COVID-19 vaccination in CPs was signed between the Government and Labor Unions (Federfarma-Assofarm) [26]. Community pharmacies were recruited on a voluntary basis, and participating pharmacists received mandatory, specific, online training courses developed by the National Institute of Health [26].

As of 9 April 2021, the number of CPs adhering to the abovementioned agreement was 11,000 out of a total of 18,000 member pharmacies (60% adherence rate), and CPs started vaccinating at different times throughout the country. In Sicily, COVID-19 vaccinations were made available in CPs from 30 August 2021, starting in the Palermo Province, and followed by other Sicilian Provinces in the upcoming months [27].

The aim of the present study was to investigate vaccine acceptance/hesitancy towards COVID-19 and influenza vaccinations, and its possible correlation with knowledge, attitudes, and practices regarding SARS-CoV-2 infection and preventive strategies, in a representative sample of community pharmacists operating in the Palermo Province.

## 2. Materials and Methods

We conducted a cross-sectional study throughout the administration of two different questionnaires to the pharmacists of the Palermo Province Pharmacists’ Order before and after the conduction of the vaccination campaign against SARS-CoV-2 at the COVID-19 vaccination center of the Palermo University Hospital (PUH).

More specifically, following an agreement with the Sicilian health regional authority, the PUH was individuated as a dedicated vaccination center for pharmacists operating in the Palermo Province.

The first questionnaire, administered from December 2020 to February 2021, was aimed at analyzing the knowledge, attitudes, practices, and vaccine adherence/hesitancy regarding COVID-19 vaccination and influenza vaccination, before the beginning of the vaccination campaign for CPs in Sicily (on the 25 February 2021). The second questionnaire, administered in October 2021, was conceived to investigate factors associated with the discrepancy observed between the percentage of those who had expressed the desire to be vaccinated against COVID-19 during January–March 2021 (64%), and the vaccination coverage observed among the pharmacists operating in the Palermo Province (96.5%) at the end of the primary COVID-19 vaccination cycle for CPs.

The administration of the two questionnaires took place through dedicated links created on the Google Documents^®^ platform, with access reserved for CPs on the restricted area of the website of the Palermo Province Pharmacists’ Order [28].

The two questionnaires were both administered anonymously and in Italian. The purpose of the study, the methods of treatment, and the conservation and protection of personal data were explained, and an informed consent form was signed and collected.

The study population included all 1736 CPs of the Palermo Province Pharmacists’ Order. Based on a 65% estimation of willingness to be vaccinated against COVID before the first survey administration (confidence interval 3), with a 99.9% desired level of significance, the minimum study population needed was 896 CPs.

The reliability and validity of the questionnaires were evaluated in a preliminary pilot testing study conducted among 45 community pharmacists. In this study, Cronbach’s alpha was calculated and corresponded to 0.87, with an adequate reliability of the test.

The study was approved on 13 November 2020 by the Ethics Committee Palermo 1 (n. 06/2020).

### 2.1. Baseline Survey

The first questionnaire consists of 25 items distributed in five sections with the purpose of investigating:Socio-demographic aspects: age, gender, residence, work location;Anamnestic data: presence of chronic diseases and type of disease;Knowledge and attitudes regarding vaccination and VPDs: confidence in scientific research and attitudes towards the influenza vaccine and other vaccinations, adherence to seasonal influenza vaccination;Willingness to be vaccinated against SARS-CoV-2 and reasons for vaccination adherence or refusal;Knowledge and perceptions on the role of alternative treatment in combating COVID-19: hyperimmune plasma or other alternative strategies.

### 2.2. Post Vaccination Survey

The second questionnaire consists of six items aimed at exploring:Adherence to the COVID-19 vaccination campaign;Possible changes of opinion to receive the COVID-19 vaccination as compared to the first questionnaire;The main reason in changing opinion on COVID-19 vaccine administration;Possible involvement during the SARS-CoV-2 vaccination campaign;Perceptions on the need to improve knowledge and training on vaccination and VPDs;The quality of the service offered by the vaccination reference center.

### 2.3. Statistical Analysis

Data obtained were collected in a Microsoft Excel database, which was automatically filled by the Google^®^ Modules online questionnaire administration system. Data were analyzed through Epi-Info software 3.5.4.

Absolute and relative frequencies were calculated for categorical (qualitative) variables. Differences in qualitative variables were analyzed using chi-square tests (or Fisher’s exact test when appropriate).

All the variables were found to have a statistically significant association with adherence to the seasonal influenza vaccination and a willingness to receive the COVID-19 vaccination in the first questionnaire. In the univariate analysis, they were included in a multivariate backward stepwise logistic regression model. In addition, all the variables with a *p*-value ≤ 0.20 were selected in the multivariate model, to guarantee a more conservative approach.

Crude odds ratios (ORs) and adjusted ORs (adj-ORs), with their 95% confidence intervals (CIs), were calculated. The level of significance chosen was a *p*-value < 0.05 (two tailed).

## 3. Results

### 3.1. Baseline Survey (January–March 2021)

The total number of participants in the first survey was 1450, with a response rate of 83.5%. As reported in Table 1, respondents were predominantly female (64.7%), and they resided in the Palermo municipality (60%). The average age of respondents was 46 years (SD ± 15.7).

The majority of respondents operated in community pharmacies (84%; *n*. 1218), only 6% (*n*. 87) in para-pharmacies, and 10% (*n*. 145) were pharmacists operating in "other" working contexts. Twenty percent of pharmacists (*n*. 290) thought they were at low risk of contracting the virus, 44% (*n*. 638) at medium risk, and 36% (*n*. 522) at high risk.

In addition, 64% of pharmacists (*n*. 928) declared that they planned to vaccinate against SARS-CoV-2, while 36% (*n*. 522) were not willing to receive the COVID-19 vaccine. Among the pharmacists who intended to vaccinate against SARS-CoV-2, the main reasons were to protect themselves (47%) and to protect their family members (44%) (Table 1).

Among the ones that were not willing to vaccinate themselves, 58% were not afraid of contracting the virus, 34% thought the vaccine was unsafe, and 8% thought it was not a necessary vaccine. Moreover, about half of participants (*n*. 725) declared that they feared long-term side effects related to the COVID-19 vaccine.

The majority of respondents (68%; *n*. 986) correctly stated that vaccines represent the best strategy to prevent SARS-CoV-2 infection, whereas the remaining 32% (*n*. 290) of respondents stated that hyperimmune plasma (20%), or other strategies (12%), could better counteract COVID-19 (Table 1).

Personal attitudes regarding influenza and other vaccinations are reported in Table 2. The majority of respondents (92%) were confident in the efficacy of vaccination in preventing VPDs; nevertheless, only 58% of pharmacists (*n* = 841) stated that they had been vaccinated against influenza during the 2020/2021 season (Table 2).

Among vaccinated pharmacists, the majority declared that they aimed to protect themselves (65%) or their family members (17%), whereas the remaining 18% of respondents were vaccinated for work-related reasons. Otherwise, among pharmacists who were not vaccinated against influenza, 52% reported that the influenza vaccine was not necessary, and 41% declared that they were not at risk of exposure to influenza viruses.

In Table 3, the univariate and multivariate analyses of factors associated with pharmacists’ willingness to vaccinate against influenza are reported. The multivariate analysis showed that the only factor that is significantly associated with a willingness to vaccinate against influenza is considering COVID-19 vaccination to be the best strategy to counteract the SARS-CoV-2 infection (Adj-OR = 1.44 (95%CI = 1.13–1.82); *p*-value: <0.01) (Table 3).

Lastly, the results of the univariate and multivariate analyses of factors associated with pharmacists’ willingness to vaccinate against COVID-19 are shown in Table 4. In the multivariable model, factors significantly associated with a willingness to receive the COVID-19 vaccination were confidence in the capacity of vaccines to prevent VPDs (Adj-OR = 1.76 (95%CI = 1.11–2.80); *p*-value: <0.05), fear of contracting SARS-CoV-2 infection (Adj-OR = 1.50 (95%CI = 1.06–2.11); *p*-value: <0.05), considering COVID-19 vaccination to be the best strategy to counteract the SARS-CoV-2 infection (Adj-OR = 1.79 (95%CI = 1.39–2.29); *p*-value: <0.001), and adherence to influenza vaccination during the 2020/2021 season (Adj-OR = 3.25 (95%CI = 2.23–4.25); *p*-value: <0.001).

### 3.2. Post-Vaccination Survey

Table 5 shows the results of the post-vaccination questionnaire administered to pharmacists who enrolled six months after the conclusion of the COVID-19 vaccination campaign. The total number of respondents among pharmacists operating in the Province of Palermo was 1391, with a response rate of 81.6%.

All the pharmacists interviewed were vaccinated against COVID-19; the vast majority of respondents (71%) received the two-dose vaccination course at the PUH vaccination center, and 94% of them rated the quality of service from good to excellent.

A total of 10% of the pharmacists interviewed (*n*. 140) positively changed their point of view about COVID-19 vaccination as compared with their previous opinion.

The main reasons were the introduction of mandatory vaccinations (44%), fear of contracting COVID-19 disease (28%), limitations on work activities (20%), and direct involvement (8%) in the COVID-19 vaccination campaign. Overall, 30% of the pharmacists surveyed (*n*. 420) reported that they had an active role in the COVID-19 vaccination campaign. More specifically, 81% of them administered vaccines at community pharmacies, and the remaining 19% prepared vaccine doses in vaccination centers. Almost 70% of respondents expressed the need to improve their knowledge and skills regarding the COVID-19 vaccine, and more generally, for all the vaccinations included in the NVP (Table 5).

## 4. Discussion

This cross-sectional study was conceived to investigate attitudes and hesitancy for influenza and COVID-19 vaccinations among pharmacists operating in the Palermo Province. CPs can be considered as the first link between citizens and the healthcare system now more than ever because of the unprecedented emergency that the world is facing with the arrival of the SARS-CoV-2 pandemic.

Although the first COVID-19 pandemic wave in Sicily was well controlled, with low morbidity and mortality rates both in the general population and in healthcare workers (HCWs), the second COVID-19 wave initiated in mid-September 2020 put a strain on the healthcare system [29]. The first phase of this study was conducted in the midst of the second COVID-19 epidemic wave that occurred in Sicily, where an average of 1540 COVID-19 cases per day were reported [30].

Herd immunity through vaccination is a key measure to control the COVID-19 pandemic; however, vaccine hesitancy remains a public health threat, which is still common among HCWs [31].

Vaccine hesitancy is typically fueled by conspiracy theories, fake news, and social media, which may even have an affect among professionals [32,33]. Typically, vaccination hesitancy among healthcare workers results from insufficient knowledge on the safety profile of vaccines [34]. Beyond a rational “risk versus benefit” analysis, an individual’s decisions about vaccination take into account multiple variables, and should therefore be considered as a continuum, rather than a dichotomous (anti- versus pro-vaccine) belief [35]. The continuum-like nature of vaccine acceptance gives us a more accurate cross-section of vaccine hesitators, who are a more heterogeneous group than one might think.

The findings of our study, despite tangible fears and concerns, showed encouraging data about confidence in vaccines in a specific core group of pharmacists since before the vaccination campaign against SARS-CoV-2 began. Adhering to previous flu vaccine campaigns was associated with a higher likelihood of accepting the COVID-19 vaccine.

However, there were still some subcategories that were less likely to vaccinate [31]. Some of them (42%), indeed, expressed high hesitation toward seasonal influenza vaccination even during the COVID-19 era. Such attitudes have been attributed to the perception of those healthcare workers of feeling that they are healthy and can withstand minor sicknesses such as influenza [13,21].

These responses may be due to a misperception on the part of community pharmacists who feel that they are not directly involved in the care of frail patients; however, on the contrary, their role in vaccine counseling must be considered crucial for public health.

Moreover, several studies conducted in Europe confirmed the key role of HCWs in modifying knowledge, personal convictions, and choices about vaccinations among the general population [36,37,38].

A similar scenario occurred regarding their willingness to get vaccinated with the COVID-19 vaccine. Thirty-six percent of respondents did not in fact want to be vaccinated against SARS-CoV-2, as the most common concern for vaccination among pharmacists was vaccine safety, which was also the most important driver of vaccine hesitancy. Concerns for safety mainly included potential side effects, especially long-term side effects. The rapid development and Emergency Use Authorization of COVID-19 vaccines had caused concerns and distrust [39,40,41,42,43]. However, according to the latest recommendations, the Centers for Disease Control and Prevention strongly encourages HCWs to receive influenza vaccines, and considering that pharmacists tend to have a strong community influence, a positive attitude toward vaccination may play a key role on the promotion of influenza vaccination among the community [44].

As suggested by the results of the post-vaccination survey, it is likely that the achievement of a much higher coverage than expected is related both to an increased fear of the virus and its clinical consequences, and especially to the strong limitations at work due to the introduction of mandatory vaccinations for HCPs in Italy and in Sicily [45].

At the same time, one third of pharmacists were more aware of their fundamental role in vaccine preparation, counseling, and administration of COVID-19 vaccines from August 2021 [27].

The global health crisis related to the COVID-19 pandemic has placed the issue of the legitimacy of imposing compulsory vaccination mandates at the center of a multifaceted debate on pandemic health policies [29]. Italy was the first European nation to introduce mandatory COVID-19 vaccinations, and the imposition of compulsory health treatments has always been a subject of animated legal and bioethical debate [45]. This choice of compulsory vaccination for HCPs demonstrated good effectiveness in achieving the highest possible degree of vaccination coverage among health professionals, and as reported in our study, among the pharmacists operating in the Province of Palermo.

This strategy was inevitable to guarantee the continuity and the safety of health systems, and the protection of the health of the patients during this pandemic [46].

Moreover, influenza and COVID-19 vaccines have proven to be extraordinarily effective tools for containing the spread of the infections, and limiting hospitalizations and deaths [47,48]. Their safety and efficacy have been widely proven in studies carried out all over the world [49]. As for adverse events, although their existence is undeniably documented, it is impossible to imagine that a worldwide vaccination campaign could result in an absolute absence of undesirable effects [50].

It is also of interest that the need emerged to implement health promotion and formative interventions on the efficacy and safety of vaccines, and to include the subject of vaccinology in the curricula of degree courses in pharmacy and its post-graduate medical schools.

There are some limitations of the present study that need to be highlighted. Firstly, a possible lack of representativeness due to the limited number of participants should be considered. Secondly, the surveys were mainly conducted through online recruitment and participants were volunteers, which may introduce potential selection bias. Moreover, vaccine acceptance was collected through self-reporting; therefore, a social desirability bias should be considered as well. Due to selection bias and social desirability bias, participants may be more interested in vaccination, or may report results based on social expectations rather than their actual thoughts; thus, this may lead to an overestimation of reported COVID-19 vaccine acceptance.

## 5. Conclusions

Vaccine hesitancy among healthcare professionals is a major public health concern that was evident during past years and also during the current COVID-19 pandemic, which has a serious impact on patient choices regarding vaccination acceptance or refusal [51,52,53,54].

Although seasonal influenza vaccinations appeared to appeal more to healthcare professionals during the COVID-19 pandemic, it is still far from the ideal scenario and from desirable vaccination coverage rates [55,56].

Moreover, the achievement of very high COVID-19 vaccination coverage rates was mainly due to the introduction of mandatory vaccinations; nevertheless, the results of our study, despite tangible fears and concerns, showed encouraging data about confidence in the COVID-19 vaccine, and other vaccines, since the first survey.

It is encouraging to discover that a willingness for COVID-19 vaccination was relatively high among pharmacists before the beginning of the vaccination campaign, and that the vast majority of pharmacists subsequently received the vaccination.

Lastly, the results of this cross-sectional study could encourage pharmacists and other HCPs to re-evaluate their own vaccination profiles, as well as those of their patients, in accordance with vaccinations recommended in the Italian national immunization plan.

## Figures and Tables

**Table 1 vaccines-10-00475-t001:** Socio-demographic characteristics, knowledge, attitudes, and perceptions regarding COVID-19 of the pharmacists recruited in the baseline survey (*n* = 1450).

	*n* (%)
**Gender**	
Male	511 (35.3)
Female	939 (64.7)
**Mean age ± SD, in years**	46.3 ± 15.7
**Residence**	
Palermo	870 (60)
Municipalities in the Palermo province	493 (34)
Other municipalities	87 (6)
**Workplace**	
Community pharmacy	1218 (84)
Para-pharmacy	87 (6)
Other	145 (10)
**Personal risk of contracting severe COVID-19 disease**	
Low	290 (20)
Middle	638 (44)
High	522 (36)
**Willingness to receive COVID-19 vaccination**	
Yes	928 (64)
No	522 (36)
**If yes, please specify the main reason**	
Protect themselves	436 (47)
Protect their families	408 (44)
Working reasons (high risk of exposure)	84 (9)
**If no, please specify the main reason**	
Limited risk of exposure to COVID-19	302 (57.8)
Vaccination considered not safe	176 (33.7)
Vaccination considered not necessary	44 (8.5)
**Fear of COVID-19 vaccination’s long-term side effects**	
Yes	725 (50)
No	725 (50)
**Best strategy in preventing SARS-CoV-2 infection**	
Vaccination	986 (68)
Hyperimmune plasma	290 (20)
Other strategies	174 (12)

**Table 2 vaccines-10-00475-t002:** Knowledge, attitudes, and perceptions towards anti-influenza vaccination (*n* = 1450).

	*n* (%)
**Vaccine confidence in the prevention of VPDs**	
Yes	1334 (92)
No	116 (8)
**Adherence to influenza vaccination**	
Yes	841 (58)
No	609 (42)
**If yes, please specify the main reason**	
Protect themselves	547 (65)
Protect their families	143 (17)
Working reasons	151 (18)
**If no, please specify the main reason**	
Unnecessary vaccination	317 (52)
No exposure to influenza	249 (41)
Unsafe vaccination	43 (7)

**Table 3 vaccines-10-00475-t003:** Factors associated with influenza vaccination adherence in the univariable (crude OR) and multivariable (adj OR) analyses among study participants in the baseline survey (*n* = 1450) (95% CI: 95% confidence intervals).

	Influenza Vaccination Adherence (Yes vs. No)			
	Crude OR (95% CIs)	*p*-Value	Adj-OR (95% CIs)	*p*-Value
**Gender**				
Female	Reference	0.85		
Male	1.02 (0.82–1.25)			
**Age category**				
≤50 years	Ref	0.40		
≥51 years	1.09 (0.88–1.35)			
**Residence**				
Palermo Province/Outside Province mun.	Ref	0.24		
Palermo municipalities	1.28 (0.85–1.95)			
**Vaccine confidence in preventing VPDs**				
No	Ref	<0.05	Ref	0.55
Yes	1.49 (1.03–2.16)		1.14 (0.74–1.74)	
**Fear of contracting SARS-CoV-2**				
No	Ref	0.18	Ref	0.62
Yes	1.22 (0.91–1.62)		0.92 (0.66–1.28)	
**Personal risk of contracting severe COVID-19 disease**				
Low/medium	Ref	0.11	Ref	0.86
High	1.23 (0.95–1.60)		1.02 (0.76–1.37)	
**Best strategy in preventing COVID-19 disease**				
Hyperimmune plasma/other strategies	Ref	<0.001	Ref	<0.01
Vaccination	1.52 (1.23–1.88)		1.44 (1.13–1.82)	
**Willingness to receive COVID-19 vaccination**				
No	Ref	0.45		
Yes	1.08 (0.88–1.37)			

**Table 4 vaccines-10-00475-t004:** Factors associated with willingness to receive COVID-19 vaccination in the univariable (crude OR) and multivariable (adj OR) analyses among study participants in the baseline survey (*n* = 1450) (95% CI: 95%confidence intervals).

	Willingness to Receive COVID-19 Vaccination (Yes vs. No)			
	Crude OR (95% CIs)	*p*-Value	Adj-OR (95% CIs)	*p*-Value
**Gender**				
Female	Reference	0.50		
Male	1.07 (0.86–1.34)			
**Age category**				
≤50 years	Ref	0.47		
≥51 years	0.92 (0.74–1.15)			
**Residence**				
Palermo Province/Outside Province mun.	Ref	<0.001	Ref	0.18
Palermo municipalities	2.60 (1.71–3.96)		1.28 (0.87–2.64)	
**Vaccine confidence in preventing VPDs**				
No	Ref	<0.001	Ref	<0.05
Yes	2.71 (1.87–3.93)		1.76 (1.11–2.80)	
**Fear of contracting SARS-CoV-2 infection**				
No	Ref	<0.001	Ref	<0.05
Yes	2.34 (1.75–3.14)		1.50 (1.06–2.11)	
**Personal risk of contracting severe COVID-19 disease**				
Low	Ref	<0.001	Ref	0.20
High	1.88 (1.44–2.45)		1.21 (0.90–1.64)	
**Fear of side effects of COVID-19 vaccine**				
Yes	Ref	0.11	Ref	0.26
No	1.18 (0.96–1.46)		1.15 (0.92–1.43)	
**Best strategy in preventing SARS-CoV-2**				
Hyperimmune plasma/Other strategies	Ref	<0.001	Ref	<0.001
Vaccination	2.30 (1.85–2.87)		1.79 (1.39–2.29)	
**Influenza vaccination adherence during 2020/2021**				
No	Ref	<0.001	Ref	<0.001
Yes	3.63 (2.56–4.89)		3.25 (2.23–4.25)	

**Table 5 vaccines-10-00475-t005:** Knowledge, attitudes, and perceptions regarding COVID-19 vaccination in the supplementary survey administered during October 2021 (*n* = 1391).

	*n* (%)
**Adherence to COVID-19 vaccination**	
Yes	1391 (100)
No	0 (0)
**Vaccination at the Hub of the University Hospital of Palermo**	
Yes	992 (71.3)
No	399 (28.7)
**Satisfaction level of the Hub of the University Hospital of Palermo**	
Excellent	500 (35.9)
Very good	671 (48.3)
Good	140 (10)
Acceptable	60 (4.4)
Unacceptable	20 (1.4)
**Opinion change towards COVID-19 vaccination in the last 6 months**	
Yes	140 (10)
No	1251 (90)
**If opinion changed, what is the main reason?**	
Mandatory vaccination for HCPs	90 (64.3)
Fear of contracting COVID-19 disease	39 (27.8)
Being involved into the vaccination campaign	11 (7.9)
**Involvement in COVID-19 vaccination campaign**	
Yes	420 (30)
No	971 (70)
**If involved, in what capacity?**	
Vaccine preparation and administration in community pharmacies	340 (80.9)
Vaccine preparation at vaccination hubs of the Palermo Province	80 (19.1)
**Need to improve competences for COVID-19 vaccines**	
Yes	841 (69.5)
No	550 (30.5)
**Need to improve competences for vaccinations in the National Vaccination Plan**	
Yes	951 (68)
No	440 (32)

## Data Availability

Data available on request due to privacy restrictions.

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
