# Peer review of "Knowledge, Attitudes, Perceptions and Vaccination Acceptance/Hesitancy among the Community Pharmacists of Palermo’s Province, Italy: From Influenza to COVID-19"

_vaccines, 2022, doi:10.3390/vaccines10030475_

Round 1

Reviewer 1 Report

The article written by Costantino et al looks to me a descent work but need several changes before a reader can understand well.

COVID-19 is expressed in two forms within title, it is misleading. Need to use consistent form of term COVID-19 or SARS-CoA-2.

Authors were comparing the responses that were taken at different time, they need to clarify whether same samples participated in both surveys? Was it a longitudinal study?

Only descriptive analysis was mentioned in the abstract. Authors should give a highlight of inference drawn from the study with the help of inferential analysis and briefly should include a note in abstract.

The abbreviation HCP suddenly appear in abstract without its expansion anywhere before.

The conclusion is recommendation here, rather than a message based on the outcome of the study. Need to be refined.

Line 76 specifies about training given to pharmacist, why still authors insist to give training in abstract section.

The aim of the study is not clear to me, they wanted to check KAP towards SARS-CoV-2, then shifted to acceptance/hesitancy towards COVID-19 and influenza. What is the connection between the two?

As per author, study was approved in November 2020 by ethics committee, was it approved for two time questionnaire administration?

How authors decided about the time of administration of questionnaire, first time between December 2020 to feb 2021, why this period was selected and what about October 2021? Why they selected this time?

How questionnaire was developed? There is no explanation of reliability of the questionnaire and its validation.

Lines 110 to 134 need to be rewritten and re-arranged. The itemization is not done properly. Read carefully.

Which language questionnaire was used?

What was the education level of the pharmacist? Was their any variation checked and their experience?

There is a need to include separately conclusion section in the text.

Author Response

Reviewer #1

Comment: The article written by Costantino et al looks to me a descent work but need several changes before a reader can understand well.

A: Dear reviewer,

Thank you for the opportunity to revise our manuscript and for appreciating the original article “Knowledge, attitudes, perceptions and vaccination acceptance/hesitancy among the community pharmacists of Palermo’s Province, Italy: from influenza to COVID-19” submitted to Vaccines.

Your useful suggestions were fully considered with attention and a point by point answer to your remarks and questions is reported below.

C: COVID-19 is expressed in two forms within title, it is misleading. Need to use consistent form of term COVID-19 or SARS-CoA-2.

A: The correction was carried out and the title was rewritten.

C: Authors were comparing the responses that were taken at different time, they need to clarify whether same samples participated in both surveys? Was it a longitudinal study?

A: The question is quite correct. The respondents to first and second questionnaires were not paired. Only the study population (the members of the Palermo’s Order or Pharmacists) was the same but the two samples (also if similar in numerosity and % of respondents) were different. The detail was specified in methods section. I confirm that is a cross-sectional study with two different questionnaires administered in two different times.

C: Only descriptive analysis was mentioned in the abstract. Authors should give a highlight of inference drawn from the study with the help of inferential analysis and briefly should include a note in abstract. The abbreviation HCP suddenly appear in abstract without its expansion anywhere before.

A: Thank you for your suggestion, the abstract was corrected mentioning the results of the inferential analysis and with the definition of healthcare professionals.

C: The conclusion is recommendation here, rather than a message based on the outcome of the study. Need to be refined.

A: As correctly suggested, the conclusion in the abstract were modified.

C: Line 76 specifies about training given to pharmacist, why still authors insist to give training in abstract section.

A: The observation is correct. The two sentences are clarified. The mandatory training for CPs in Italy was dedicated only to COVID-19 vaccination. At the same time, a need of training for all the vaccination included in the National Vaccination Plan emerged from the second survey.

C: The aim of the study is not clear to me, they wanted to check KAP towards SARS-CoV-2, then shifted to acceptance/hesitancy towards COVID-19 and influenza. What is the connection between the two?

A: Thank you for your comment. The objectives of the study were further clarified. Specifically, the main objective was to evaluate vaccine acceptance/hesitancy against COVID-19 and influenza, analyzing with inferential analysis any correlation with KAP of these two diseases and preventive strategies.  Unfortunately, among vaccinated women we did not ask the main reason for vaccination adherence.

C: As per author, study was approved in November 2020 by ethics committee, was it approved for two time questionnaire administration?

A: Yes, the study was approved for all the items in the two questionnaire, that was shifted because some of the questions (specifically those contained in the second one) were mainly dedicated to a post-vaccination offer survey.

C: How authors decided about the time of administration of questionnaire, first time between December 2020 to feb 2021, why this period was selected and what about October 2021? Why they selected this time?

A: Thank you for your observation. The first questionnaire was administered during the first phase of HCP vaccination in Italy mainly dedicated to hospital and long-term care facilities employees (started the 27th of December 2020) and before the beginning of vaccine administration to territorial HCPs (such as CPs), that started the 25th of February 2021. The second survey was administered before the beginning of the third dose administration and when the primary cycle vaccination finished and the data on vaccination coverage of the Palermo’s Order of Pharmacists were consolidated (96.5% as reported in the text in material and methods).

C: How questionnaire was developed? There is no explanation of reliability of the questionnaire and its validation.

A: We perfectly agree with your observation. An explanation of the reliability and validation of the questionnaire was added.

C: Lines 110 to 134 need to be rewritten and re-arranged. The itemization is not done properly. Read carefully.

A: Thank you for your correct observation. The itemization was corrected.

C: Which language questionnaire was used?

A: Italian language

C: What was the education level of the pharmacist? Was their any variation checked and their experience?

A: The education level is quite uniform because in Italy CPs need the degree in Pharmacy or in Chemistry and Pharmaceutical Technology and also be enabled to the profession to work in a Pharmacy. So the educational level was not evaluated and compared with independent variable.

C: There is a need to include separately conclusion section in the text.

A: As correctly suggested the conclusion section was separated from discussion section. 

Reviewer 2 Report

I was invited to revise the paper entitled "Knowledge, attitudes and perceptions towards SARS-CoV-2 and vaccination acceptance/hesitancy among the community pharmacists of Palermo’s Province, Italy: from influenza to COVID-19". It was a cross-sectional study made by two different questionnaires aimed to assess the attitudes towards Covid vaccination among pharmacists from a Province of Souther Italy.

The topic is interesting and can improve the knowledge for the field. I have some minor observations:

  • Sample size estimation is lacking;
  • It is unclear if questionnaires data were paired. If yes, it should be useful to analyze the change in attitude between two study periods;
  • Aithgors should also analyze factor associated to the involvement into COVID-19 vaccination campaign among pharmacists;
  • As supplementary analysis, Authors should analyze the association between age category and flu-vaccine and covid-vaccine acceptance;
  • Authors should compare their results with previous literature;
  • Strenght and limitation section is missing.

Author Response

Reviewer #2

Comment: I was invited to revise the paper entitled "Knowledge, attitudes and perceptions towards SARS-CoV-2 and vaccination acceptance/hesitancy among the community pharmacists of Palermo’s Province, Italy: from influenza to COVID-19". It was a cross-sectional study made by two different questionnaires aimed to assess the attitudes towards Covid vaccination among pharmacists from a Province of Southern Italy.

The topic is interesting and can improve the knowledge for the field. I have some minor observations.

A: Dear reviewer,

Firstly, thank you for the opportunity to revise our manuscript and for appreciating the article “Knowledge, attitudes, perceptions and vaccination acceptance/hesitancy among the community pharmacists of Palermo’s Province, Italy: from influenza to COVID-19” submitted to Vaccines.

Thank you for the opportunity to revise our manuscript. We hope that this revised version could improve the article and may have solved the minor observation raised in your revision.

Your useful comments were considered with attention and a point by point answer to your remarks and questions was reported below.

C: Sample size estimation is lacking;

A: Thank you for your useful suggestion. We have specified the sample size estimation in material and methods section.

C: It is unclear if questionnaires data were paired. If yes, it should be useful to analyze the change in attitude between two study periods;

A: Thank you for the request. The  two surveys were not paired. The change in attitude between the study period was only estimated with aggregate data and reported in  3.2 section of Results.

C: Aithgors should also analyze factor associated to the involvement into COVID-19 vaccination campaign among pharmacists;

A: Thank you for suggestion. The involvement into COVID-19 campaign among pharmacists is a factor that was evaluated only in the post intervention survey (the participation to vaccination campaign of CPs started in Sicily only in the month of August 2021. Since that the two questionnaire were not paired (the samples were both anonymous), unfortunately this factor could not be included in the multivariable analysis.

C: As supplementary analysis, Authors should analyze the association between age category and flu-vaccine and covid-vaccine acceptance

A: Thank you for your suggestion. As reported in table 3 and 4 the age category was analyzed and the p-values obtained against flu (0.40) and COVID-19 (0.49) did not allow to insert the variable in the multivariable analysis.

C: Authors should compare their results with previous literature

A: Following your useful suggestion, in the discussion section a further comparison with literature data was implemented

C: Strenght and limitation section is missing

A: As correctly suggested we described limitations of the study in the final paragraph of discussion section.

Round 2

Reviewer 2 Report

Authors addressed all comments and the paper is now acceptable for publication.